# Towards universal health coverage: The level and determinants of enrollment in the Community-Based Health Insurance (CBHI) scheme in Ethiopia: A systematic review and meta-analysis

Aklilu Habte[1]*, Aiggan Tamene[1], Tekle Ejajo[1], Samuel Dessu[2], Fitsum Endale[1], Addisalem Gizachew[1], Dawit Sulamo[1]

1 School of Public Health, College of Medicine and Health Sciences, Wachemo University, Hosanna, Ethiopia, 2 Department of Public Health, College of Medicine and Health Sciences, Wolkite University, Wolkite, Ethiopia

* akliluhabte57@gmail.com

## Abstract

### Background

Community-based health insurance (CBHI) is a risk-pooling approach that tries to disperse health expenditures across families with varying health profiles to provide greater access to healthcare services by allowing cross-subsidies from wealthy to poor populations. It is crucial to assess the level of CBHI enrolment and its determinants in Ethiopia, where government health spending is limited to less than 5% of GDP, far below the Alma Ata Declaration's benchmark of 15%. Although various epidemiological studies on CBHI enrolment status and its determinants have been undertaken in Ethiopia, the results have been inconsistent, with significant variability. However, no nationwide study assessing the pooled estimates exists today. Furthermore, the estimated strength of association at the country level varied and was inconsistent across studies. Hence, this systematic review and meta-analysis aimed at estimating the pooled prevalence of CBHI enrolment and its determinants in Ethiopia.

### Methods

A comprehensive search of studies was done by using PubMed, EMBASE, Science Direct, HINARI, Scopus, Web of Science, and the Cochrane Library. The database search was complemented by google scholar and some repositories for grey literature. The search was carried out from February 11 to March 12, 2022. The relevant data were extracted using a Microsoft Excel 2013 spreadsheet and analyzed using STATA™ Version 16. Studies reporting the level and determinants of CBHI enrolment in Ethiopia were considered. A weighted DerSimonian Laired random effect model was applied to estimate the pooled national prevalence of CBHI enrolment. The Cochrane Q test statistics and $I^2$ tests were used to assess

**Data Availability Statement:** All relevant data are within the article and its Supporting Information files.

**Funding:** The author(s) received no specific funding for this work.

**Competing interests:** The authors have declared that no competing interests exist.

**Abbreviations:** CBHI, Community-based health insurance; FMOH, Federal Ministry of Health; HSTP, Health Sector Transformation Plan; JBI, Joanna Briggs Institute; LMICs, Low and Middle-Income Countries; OOP, Out-of-pocket; OR, Odds Ratio; PRISMA, Preferred Reporting Items for Systematic Reviews and Meta-Analysis; SNNPR, South Nations and Nationalities People of the Region; UHC, Universal Health Coverage; WHO, World Health Organization.

the heterogeneity of the included studies. A funnel plot, Begg's and Egger's tests, were used to check for the presence of publication bias.

## Results

Fifteen studies were eligible for this systematic review and meta-analysis with a total of 8418 study participants. The overall pooled prevalence of CBHI enrolment in Ethiopia was 45.5% (95% CI: 32.19, 58.50). Affordability of premium for the scheme[OR = 2.58, 95% CI 1.68, 3.47], knowledge of respondents on the CBHI scheme[OR = 4.35, 95% CI 2.69, 6.01], perceived quality of service[OR = 3.21, 95% CI 2.04, 4.38], trust in the scheme[OR = 2.32, 95% CI 1.57, 3.07], and the presence of a person with a chronic disease in the household [OR = 3.58, 95% CI 2.37, 4.78] were all found to influence CBHI enrolment.

## Conclusion

Community health workers (CHWs) need to make a high effort to improve knowledge of CBHI in rural communities by providing health education. To deal with the issue of affordability, due emphasis should be placed on building local solidarity groups and strengthening local initiatives to aid poor members. Stakeholders in the health service delivery points need to focus on the dimensions of high service quality. The financial gap created by the adverse selection of households with chronically ill members should be rectified by implementing targeted subsidies with robust plans.

## Introduction

Globally, about 44 million households face catastrophic healthcare spending, while approximately 25 million households are impoverished as a result of these direct healthcare expenses [1]. Since many governments have failed to establish formal financial protection mechanisms and insurance schemes, the majority of the world's poor must rely on their resources to pay for medical expenses [2, 3]. Low and middle-income countries (LMICs) account for about 90% of these cases [1]. This direct healthcare expenditure accounted for 37% in Ethiopia [70], and out-of-pocket (OOP) expenditure has risen over time and remains one of the country's major hurdles to healthcare access [4].

United Nations formulated a list of 17 Sustainable Development Goals (SDGs) in 2014, with the third goal being to "ensure healthy lives and promote well-being for all at all ages" [5]. The attainment of Universal Health Coverage(UHC) is a key element of this target 3.8 [6]. UHC is a system in which "all individuals and communities have access to the health care they need without incurring financial hardship" [7, 8]. Any UHC strategy must deal with the issues of healthcare financing, access to safe, effective, high-quality basic healthcare services, resource distribution, and financial hardship protection [9]. Reforms like community-based health insurance (CBHI), are vital to ensuring that health systems contribute to health equity, and social justice, primarily by advancing toward universal access and social health protection [10]. Ethiopia is striving to achieve UHC through state-managed schemes such as CBHI [11].

CBHI is a risk-pooling approach that aims to disperse health expenditures across families with varying health profiles to provide greater access to healthcare services by allowing cross-subsidies from wealthy to poor populations [12, 13]. It is an autonomous, not-for-profit, voluntary, and member-based scheme in which the community is actively involved in driving its setup and management [3, 14]. It is a non-profit health-financing mechanism based on the

principles of solidarity and risk-sharing [12]. There is substantial evidence that CBHI increases resource mobilization that improves health service utilization and provides financial security for members by lowering OOP expenses [15]. Studies have demonstrated that enrollment in the CBHI scheme has a significant impact on the utilization of maternal, neonatal, and child health services [16–18].

Ethiopia has been piloting the CBHI program in 13 pilot districts since 2011, intending to learn from the experience, and eventually, it is expanding across the country [12]. The number of schemes had expanded to 377 districts by the end of June 2017 [19]. Meanwhile, evaluations of the original pilot schemes have found that CBHI had a mostly positive impact on increasing healthcare uptake by decreasing financial hardship [19]. The ultimate goal of Ethiopia's CBHI scheme is to ensure that all Ethiopian families have equitable and sustainable access to high-quality health care, by raising financial stability and encouraging social inclusion through the health sector [12].

Despite growing support for the adoption of CBHI, empirical evidence suggests that enrollment in the scheme has remained low in areas where it has already been implemented [70]. This low enrollment rate had a negative influence on healthcare accessibility [20]. The low implementation level has been attributed to several interconnected factors at the individual, institutional, and policy levels [21, 22]. These dimensions are described as household, scheme-related, institutional, and supply-side factors [10, 23]. Poor service satisfaction, low perceived quality of care, lack of trust in the scheme, and provider incompetence could all be factors contributing to low scheme uptake in Ethiopia [10, 19]. Failure to understand the risk pooling principle, a lack of knowledge, and a lack of systematic integration of the social context were found to be hurdles to enrollment in status [24–26].

Enrollment in the CBHI scheme has been greatly varying across regions and districts of Ethiopia due to more recent political and social conditions [11]. This low enrolment status, along with the high cost of services, results in low annual health service utilization rates of 0.36 contacts per person which resulted in an overall poor health outcome [12]. On the other hand, the economic burden of direct OOP payments increased among the poor, who struggle to meet their daily food and shelter expenditures, leading to borrowing and asset sales [27]. The Ethiopian Ministry of Health (MOH) envisions all of its citizens having equal and affordable access to health services through the strengthening of a CBHI scheme across the country as part of its Health Sector Transformation Plan (HSTP) [28].

To assure UHC through CBHI, countries must find a means to make enrolment compulsory for their citizens and strengthen the professionalization of CBHI management with community participation [8]. It is crucial to assess the level of CBHI enrolment and its determinants in Ethiopia, where government health spending is limited to less than 5% of GDP, far below the Alma Ata Declaration's benchmark of 15% [29]. Although various epidemiological studies on CBHI enrolment status and its determinants have been undertaken in Ethiopia, the results have been inconsistent, with significant variability. However, no nationwide study assessing the pooled estimates exists today. Furthermore, the estimated strength of association at the country level varied and was inconsistent across studies. Such disparities in prevalence and determinants may not be satisfactory for policymakers and program planners to intervene in the problem of low enrolment, demanding an assessment of the pooled prevalence. Hence, this systematic review and meta-analysis aimed at estimating the pooled prevalence of CBHI enrolment and its determinants in Ethiopia. Knowing the enrolment status and its determinants at the national level is valuable for devising actions to address challenges, by making the premium system more flexible, forming partnerships with health care facilities, and strengthening regulatory frameworks [9]. The findings of this study could assist policymakers and program planners in developing relevant interventions to accelerate the country's progress toward UHC.

## Methods

### Study design

A systematic review and meta-analysis was conducted to assess the level of CBHI enrolment and its determinants in Ethiopia. The Preferred Reporting Items for Systematic Reviews and Meta-Analysis (PRISMA) 2020 guideline was used to report this systematic review and meta-analysis [30] [**S1 File**]. Also, a PRISMA flow chart was used to describe the selection of included studies to the outcome of interest (i.e. CBHI enrolment and its determinants).

### Search strategy

A comprehensive search of published and unpublished studies was conducted by using electronic databases such as PubMed, EMBASE, Science Direct, HINARI, Scopus, Web of Science, and the Cochrane Library. The database search was supplemented by looking for unpublished studies (grey literature) using Google Scholar, Google, and institutional repositories, such as the Addis Ababa University Digital Library. The search was carried out from February 11 to March 12, 2022, and all studies published until the last day of the search (March 12, 2022), were included in the analysis. Medical Subject Heading (MeSH) and keywords were used to identify relevant studies from the respective database. In addition, Boolean operators ('OR' and 'AND') and truncation (*) were used with the keywords. The search syntax was created for PubMed initially, and subsequently modified to meet the additional database-specific search requirements (**S2 File**). The search strategy used to retrieve relevant articles in PubMed search was ("Enrolment"[All Fields] OR "enrol*"[All Fields] OR "adopt*"[All Fields] OR "Uptake"[All Fields] OR "uptak*"[All Fields] OR "using"[All Fields] OR "usage"[All Fields] OR "utiliz*"[All Fields]) AND ("community based health insurance"[MeSH Terms] OR "community based health insurance"[MeSH Terms] OR "Community Health Insurance"[All Fields] OR "health insurance"[All Fields] OR "Universal Health Insurance"[MeSH Terms]) AND ("Ethiopia"[MeSH Terms] OR "Ethiopia"[All Fields] OR "Ethiopia s"[All Fields]). Studies that may have been missed in the initial searches were accessed by searching the reference lists of previously identified studies.

### Eligibility criteria

**Inclusion criteria.** Study design: Observational (cross-sectional and case-control) studies reporting CBHI enrolment status and its determinants were considered.

*Study area*. Only studies conducted in Ethiopia were included.

*Language*. Only studies conducted in the English language were considered for the sake of clarity, and simplicity of interpretations.

*Publication status*. Published and unpublished studies were considered, and if a study was presented in multiple reports, the most complete and up-to-date version was taken.

*Publication period*. All relevant studies reported up to March 12, 2022, were included.

**Exclusion criteria.** Systematic reviews, case studies, commentaries, letters to editors, conference abstracts, qualitative investigations, and other opinion essays were excluded. Due to the difficulty of assessing methodological quality in the absence of the full text, studies that were not fully available after two emails with the primary/corresponding author were also excluded. Furthermore, papers that did not report the outcome variable and were published in a language other than English were omitted. Studies that looked at people's willingness to pay (WTP) and willingness to join (WTJ) a CBHI scheme were excluded since they didn't clearly show active CBHI membership.

## Study selection and data extraction

To identify and select relevant studies, the PRISMA reporting guideline for PICOS was followed. Initially, duplicated retrievals were removed. Articles were then assessed for inclusion by three separate reviewers(AH, AT and TE) by examining the titles and abstracts. For the remaining articles, the full text was assessed if they met the inclusion criteria. Disagreements that arose during the study selection process were resolved by discussion. Finally, eligible studies were extracted using a data extraction form prepared using a Microsoft Excel 2013. All relevant data were extracted by using a standardized data extraction spreadsheet. The data extraction sheet includes the following variables; the name of the first author, publication year, region, study area, study setting (whether it is the institution or community-based), study design, sample size, response rate, the magnitude of CBHI enrolment and determinants with odds ratio and the corresponding 95% confidence interval. Four authors (AH, AT, SD and TE) independently extracted the relevant information, and any disagreements that arose during data extraction were handled by discussion.

## Data quality

The methodological quality of the selected studies was assessed using Joana Brigg's Institute (JBI) critical appraisal checklist based on each study design [31]. These critical appraisal tools for prevalence and case-control studies have nine and ten parameters, respectively. This was performed independently by two authors (AH and FE). Each item was assessed as either low or high risk of bias. Each parameter in the risk of bias assessment tool had equal weight. The authors provided a score of '0' if the study satisfied each specific parameter and '1' if not the case. A composite quality index was computed and the risk of bias was graded as low (0–2), moderate (3 or 4), or high ($\geq$5) (**S3 File**). Articles with low and moderate risk of bias were considered for this systematic review and meta-analysis.

## Outcome measurement

Two main outcomes were measured. The primary outcome of this study was the pooled prevalence of CBHI enrolment which was taken from each primary study. CBHI enrolment is confirmed when households pay the pre-determined membership fee, which is determined by having an updated service card and becoming eligible to receive health services. The secondary outcome variable was determinants of CBHI adoption, which was computed by using an adjusted odds ratio with 95% confidence intervals.

## Data analysis

Microsoft Excel 2013 was used to extract all of the necessary data, which was then exported to STATA version 16 for analysis. The pooled prevalence of CBHI enrolment in Ethiopia was determined using a random-effects model with the DerSimonian-Laird method. In addition, the determinants of CBHI uptake were established using pooled odds ratios with 95% confidence intervals. The $I^2$ test (p-value<0.05) was used to assess the heterogeneity between the studies. Low, moderate, and high heterogeneity were considered when $I^2$ values were <50%, 50–75, and >75%, respectively. During computation, there was significant heterogeneity between the studies ($I^2$ = 99.18%, p<0.001). Subgroup analyses were conducted by geographical regions of the country and the study year. Overall pooled odds ratios were computed. Finally, forest plots were used to present the pooled estimates for CBHI enrolment and its determinants, with their corresponding 95% confidence intervals.

To determine the possible source of heterogeneity among the point estimates of the individual studies, a meta-regression analysis was conducted. Graphically publication bias was evaluated by using a funnel plot. Statistically, Begg's and Egger's tests were conducted, and a p-value < 0.05 indicated the presence of a small study effect (publication bias) [32, 33].

## Results

### Study selection

The review identified a total of 536 published and unpublished studies by searching electronic databases. Overall, 291 studies were found to be duplicated and thus removed. The remaining 245 studies were eligible for screening. Based on the title and abstract screening, 204 studies were excluded, and 41 full articles remained. Of those 41 articles, 26 were excluded (eight owing to insufficient data, eleven failed to state the outcome of interest clearly, one case report, and six were qualitative studies). Finally, 15 studies were found to be eligible for the final systematic review and meta-analysis (Fig 1).

### Characteristics of included studies

Fifteen studies (11 cross-sectional and 4 case-control) with a total of 8418 study participants were considered [34–48]. Both the minimum (n = 262) [41] and the maximum(n = 1014) [43] sample size were from the Amhara region. The studies were conducted from 2017 to 2022. Regarding the distribution of the studies across the region, five were from Amhara [40, 41, 43, 45, 47], four from Oromia [34, 35, 44, 46], two from Sidama [39, 42], one from Somali [37], one from Southern Nations, Nationalities, and Peoples' Region (SNNPR) [36], and two from Addis Ababa [38, 48]. In all of the studies, data was collected by a face-to-face interview using a pretested, interviewer-administered questionnaire (Table 1).

### The pooled prevalence of CBHI enrolment in Ethiopia

A meta-analysis of 11 studies showed that the pooled prevalence of CBHI enrolment among households in Ethiopia was 45.5% (95% CI: 32.19, 58.82). As a result of the substantial heterogeneity across the included studies ($I^2$ = 99.2, p = 0.00), a random-effect model with a DerSimonian and Laird random method was applied to estimate the pooled prevalence. The region with the highest, 77.9% (95% CI: 74.3, 81.5), and the lowest, 12.8% (95% CI: 7.8, 17.8) CBHI enrolment status, were Amhara [47], and Sidama [39], respectively (Fig 2).

### Subgroup analysis

Because there was statistically significant heterogeneity, a subgroup analysis was conducted. A subgroup analysis was done by geographical region, study period, and publication year. Heterogeneity has persisted in the subgroup analysis in two of the aforementioned parameters. Accordingly, CBHI enrollment was highest in the Amhara region [68.03% (95%CI: 55.5, 80.5)], while it was lowest in the Sidama region [16.5% (95% CI:9.25, 23,75)] **(Fig 3)**.

By looking at the year in which the studies were conducted, the level of CBHI enrolment was found to be low before 2020, 33.98% (95% CI: 14.2, 53.75), and exhibited an increase since 2020, 59.31% (95% CI: 44.15, 70.48) **(Fig 4)**.

### Heterogeneity and publication bias

A univariate meta-regression analysis was run using study-level characteristics (study year and sample size) as a cofactor to identify the possible source of variability across the included

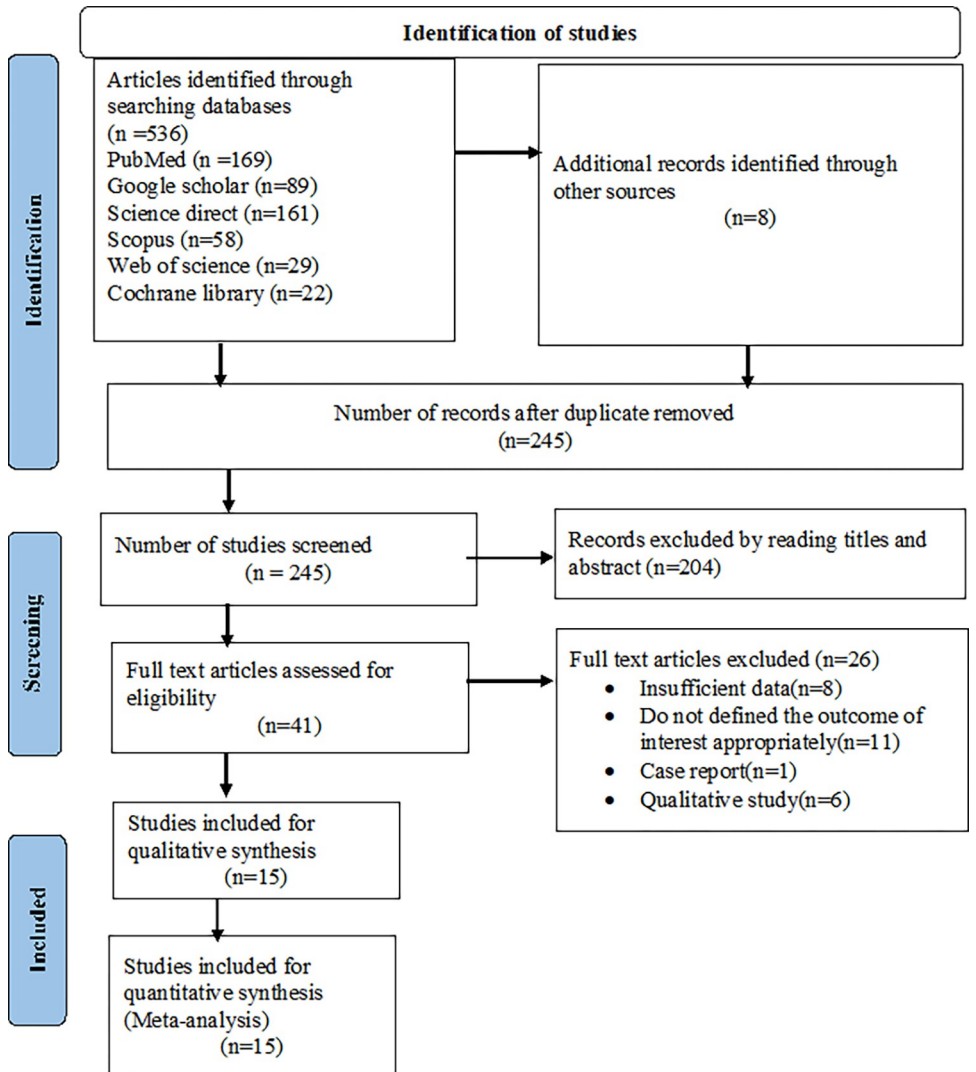

**Fig 1. PRISMA flow diagram describing the selection of studies for systematic review and meta-analysis.**

studies. However, none of these were found to be statistically significant and this indicates that between-study variability was less likely (Table 2).

The funnel plot was used to graphically examine publication bias, and the effect estimates had a symmetrical distribution, indicating that publication bias was less likely (**Fig 5**). In addition, the Egger's test was used to determine whether or not there was a small-study effect and didn't show any evidence of a small-study effect [bias coefficient = 6.48(95% CI = -36.81, 49.76); p = 0.370]. Furthermore, Begg's test was performed, and it revealed evidence of publication bias was less likely (standard error = 12.806, p-value = 0.081).

## Sensitivity analysis

To detect the impact of a single study on the overall meta-analysis estimate, a sensitivity analysis using a random-effects model was conducted. As a result, there is no evidence that a single study had an impact on the overall pooled estimate of CBHI enrolment in Ethiopia (**Fig 6**).

**Table 1. Descriptive summary of studies included in systematic review and meta-analysis of CBHI enrolment and its determinants in Ethiopia, 2017–2022.**

| Author name, year of publication | Region | Study area | Study design | Sampling techniques | Sample size | Response rate | Enrolled in the CBHI scheme | Percent | Risk of bias |
|---|---|---|---|---|---|---|---|---|---|
| Geferso et al., 2022 [34] | Oromia | Akaki | CS | srs | 600 | 97.0 | 398 | 66.3 | Low |
| Meseret, et al., 2021 [35] | Oromia | Gida Ayana | CCS | SR | 332 | 100.0 | _ | _ | Low |
| Glagn et al., 2021 [36] | SNNPR | Segen | CS | SR | 820 | 97.3 | 273 | 33.3 | Low |
| Elmi et al, 2021 [37] | Somali | Awbarre | CCS | srs | 216 | 100.0 | _ | _ | Moderate |
| Tadesse, 2021 [38] | AA | Bole Subcity | CS | SR | 435 | 99.3 | 312 | 71.2 | Moderate |
| Getasew et al., 2020 [40] | Amhara | Achefer | CCS | PPS | 296 | 100.0 | _ | _ | Low |
| Muluken et al., 2020 [41] | Amhara | Tach-Armacho | CCS | SR | 262 | 100.0 | _ | _ | Low |
| Nageso et al., 2020 [39] | Sidama | Boricha | CS | srs | 632 | 97.8 | 81 | 12.8 | Moderate |
| Bifato et al., 2020 [42] | Sidama | Daye | CS | SR | 762 | 98.9 | 154 | 20.2 | Low |
| Alebel et al., 2020 [43] | Amhara | South Gondar | CS | srs | 1014 | 98.0 | 706 | 68.17 | Low |
| Mekuria et al., 2020 [44] | Oromia | West Shoa | CS | srs | 587 | 96.2 | 130 | 22.1 | Low |
| Tsega et al., 2020 [45] | Amhara | West Gojjam | CS | srs | 690 | 97.0 | 400 | 58 | Low |
| Belay et al., 2019 [46] | Oromia | Gida Ayana | CS | SR | 631 | 98.0 | 174 | 27.5 | Low |
| Workneh et al,2017 [47] | Amhara | Tehuldere | CS | srs | 511 | 96.4 | 398 | 77.9 | Low |
| Shibeshi, 2017 [48] | AA | Sebeta | CS | SR | 630 | 99.3 | 272 | 43.2 | Moderate |
| | | | | | 8418 | 98.3 | | | |

AA: Addis Ababa, CCS: Case-control study, CS: Cross-sectional study, SRS: systematic random sampling, srs: simple random sampling, SNNPR: Southern Nations, Nationalities, and Peoples' Region

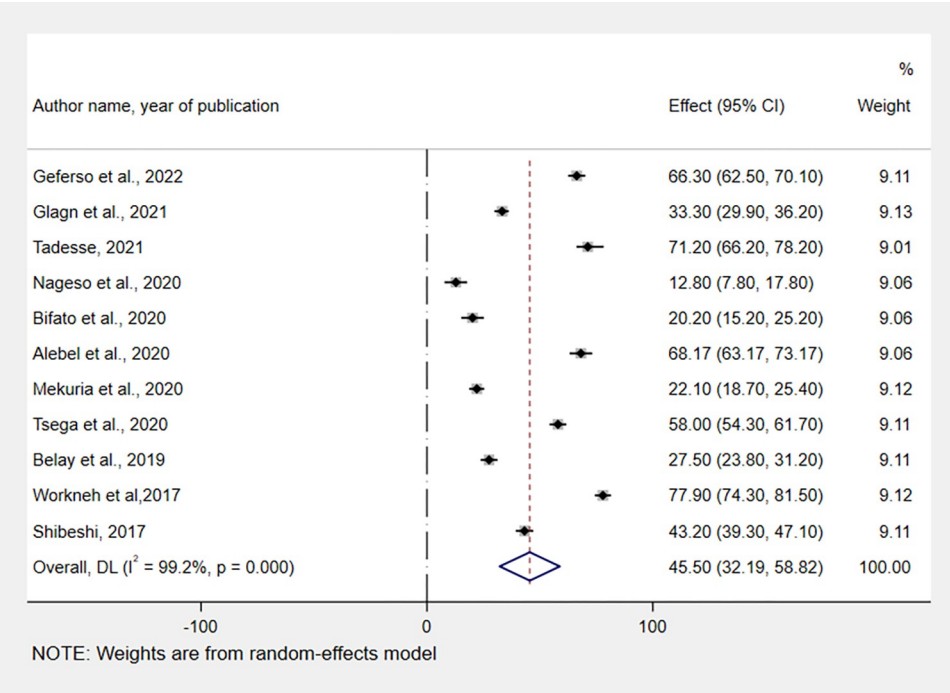

**Fig 2. Forest plot showing the pooled prevalence of CBHI enrolment in Ethiopia, 2017–2022.**

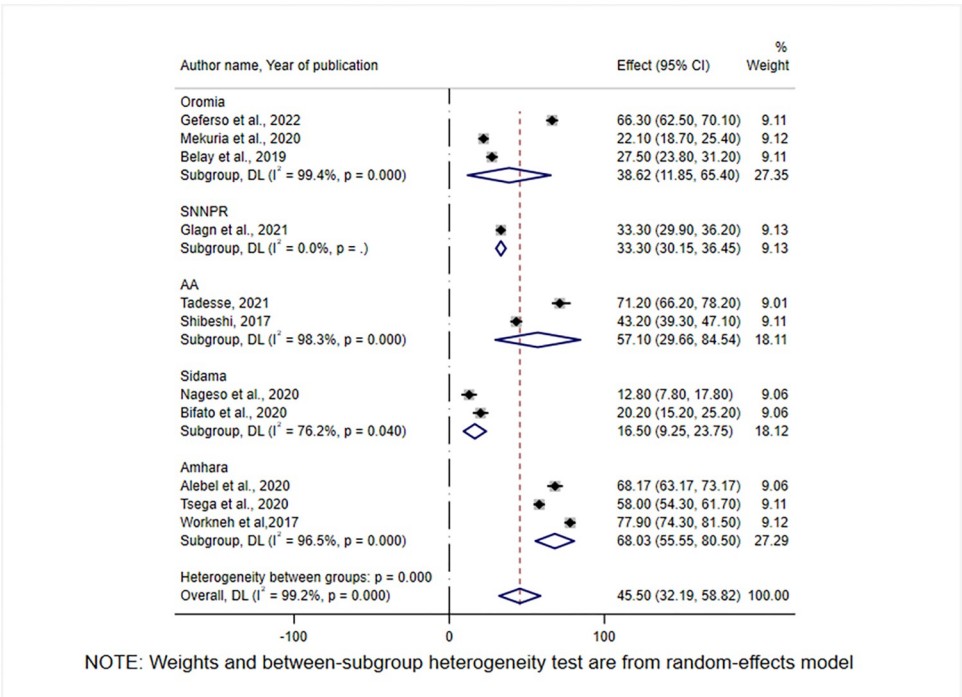

**Fig 3. Forest plot showing subgroup analysis of CBHI enrolment by geographical regions of Ethiopia, 2017–2022.**

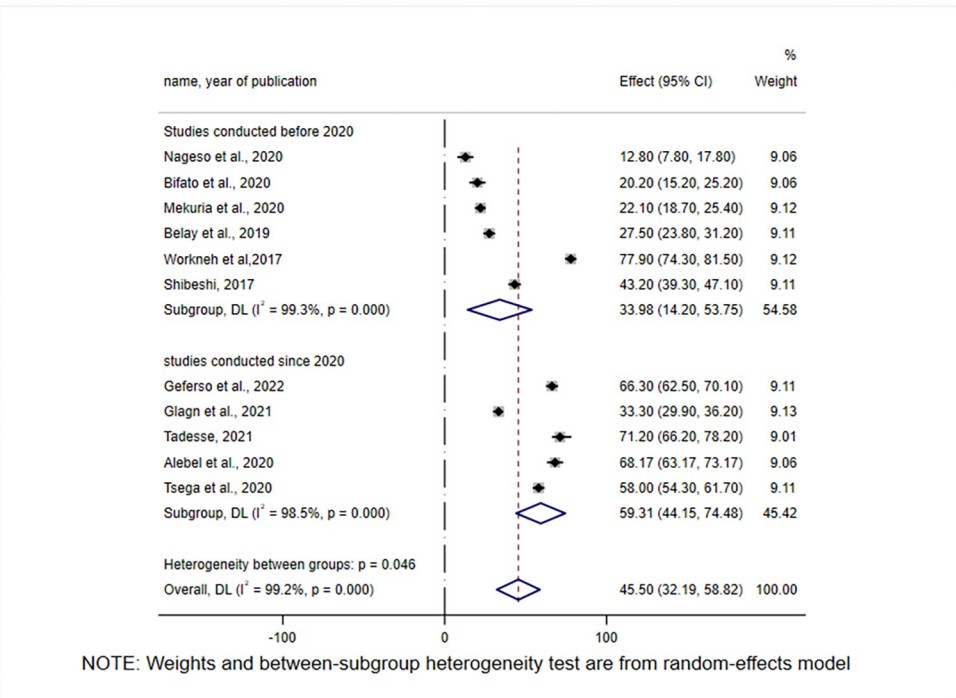

**Fig 4. Subgroup analysis of CBHI enrolment in Ethiopia by study years, 2017–2022.**

**Table 2. A univariate meta-regression analysis of factors affecting between-study heterogeneity.**

| Heterogeneity source | Coefficients | Std.Err | p-value |
|---|---|---|---|
| Sample size | -.0194809 | 0.0487589 | 0.689 |
| Publication year | -1.22279 | 4.987468 | 0.803 |

### Determinants of enrolment in the CBHI scheme in Ethiopia

To identify determinants of CBHI enrolment in Ethiopia, twelve factors were extracted from the included studies (**S4 File**). Five variables were found to be determinants of CBHI enrolment: affordability of payment for the scheme, knowledge of respondents on the CBHI scheme, perceived quality of service, trust in the scheme, and presence of a person with chronic disease in the household.

The effect of affordability (people's ability to pay) for CBHI enrollment was assessed using the findings of five studies [35, 37, 42, 46, 47]. Compared to their counterparts, individuals who thought that the CBHI premium payment was affordable were 2.58 times more likely to enrol in the scheme (OR = 2.58, 95% CI: 1.68, 3.47) (Fig 7).

A total of nine studies were included to identify the association between the knowledge of the respondents on and their enrolment in the CBHI scheme [36, 37, 42, 43, 45–48]. According to the random-effect meta-analysis, respondents with adequate knowledge of the CBHI scheme were 4.35 times more likely to enroll in the CBHI scheme than those with inadequate knowledge(OR = 4.35, 95% CI: 2.69, 6.01) (Fig 8).

Five studies [35, 39, 42, 43, 45] were eligible and considered to examine the impact of perceived service quality on enrolment in the CBHI scheme. Respondents who had a positive

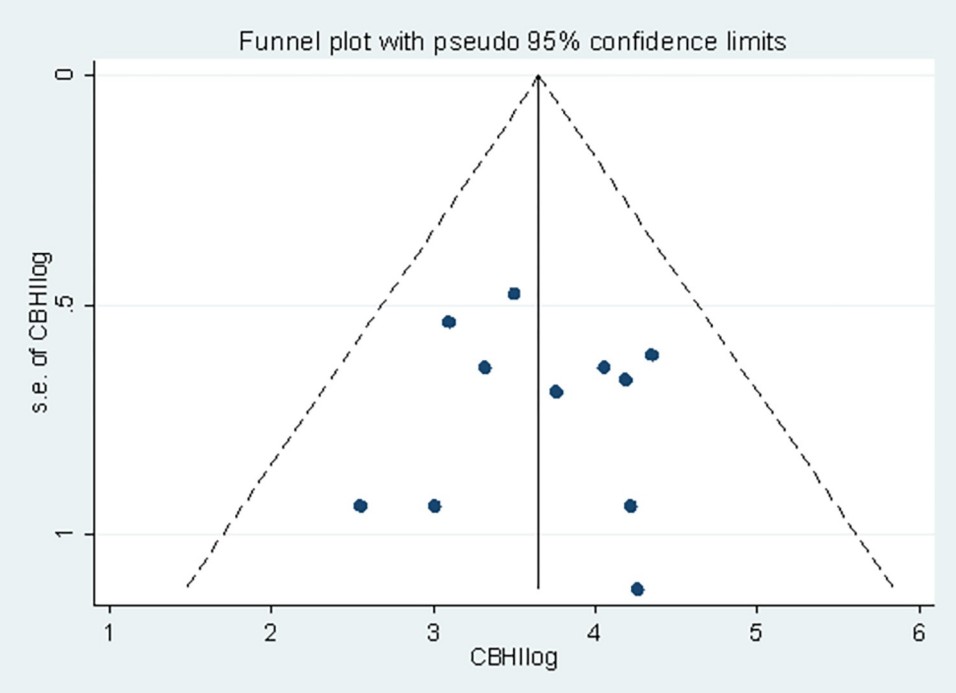

**Fig 5. Funnel plot displaying publication bias of studies reporting the level of CBHI enrolment in Ethiopia, 2017–2022.**

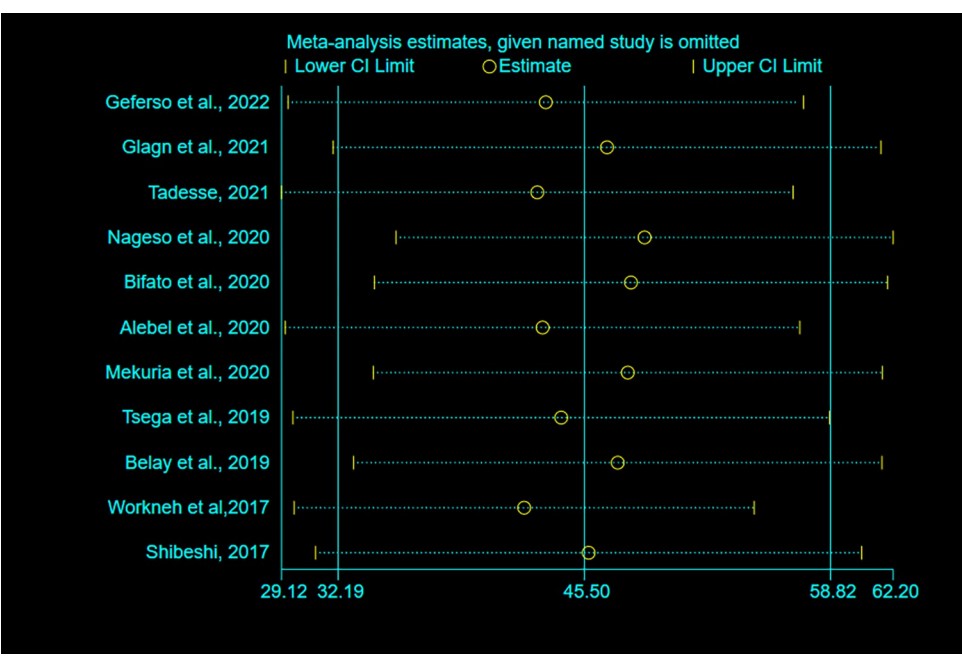

**Fig 6. Sensitivity analysis for pooled estimates of CBHI enrollment in Ethiopia, 2017–2022.**

perception of the quality of health services provided by health facilities were three times more likely to enroll in the CBHI scheme than those who had a negative perception of health service quality (OR = 3.21, 95% CI: 2.04, 4.38) (Fig 9).

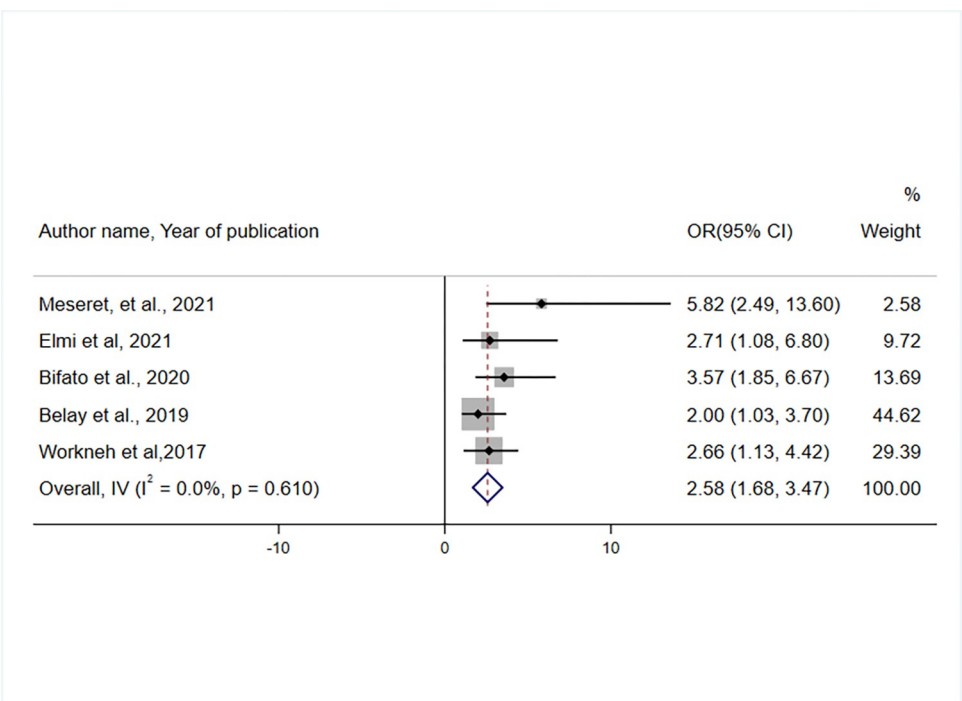

**Fig 7. Forest plot showing an association between perceived affordability of premium payment and CBHI enrolment in Ethiopia, 2017–2022.**

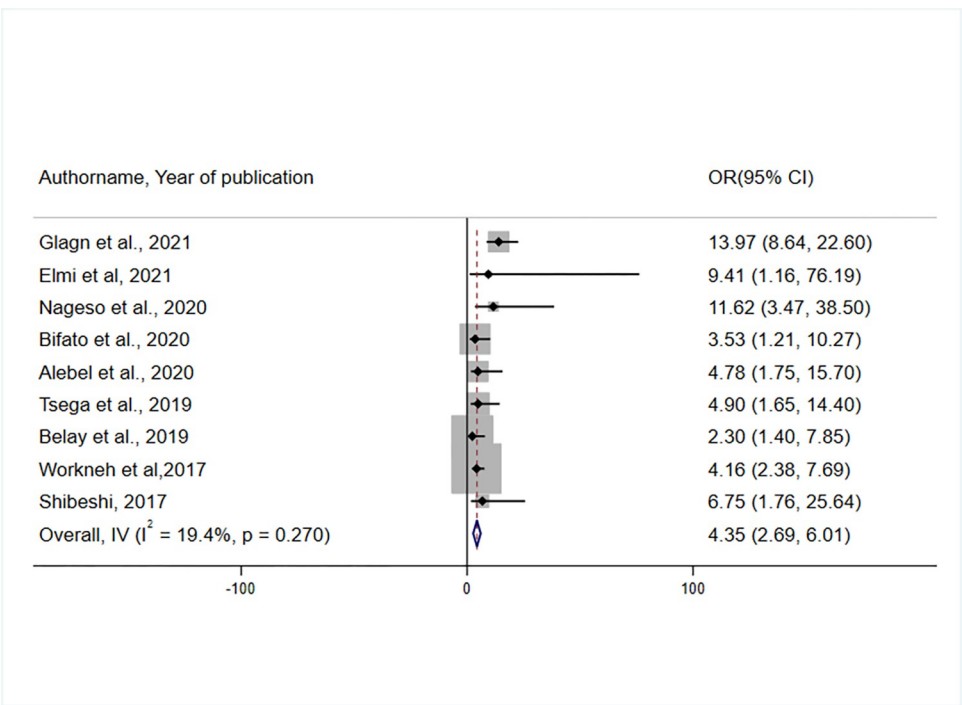

**Fig 8. Forest plot showing the association between knowledge of CBHI scheme in CBHI enrolment in Ethiopia, 2017–2022.**

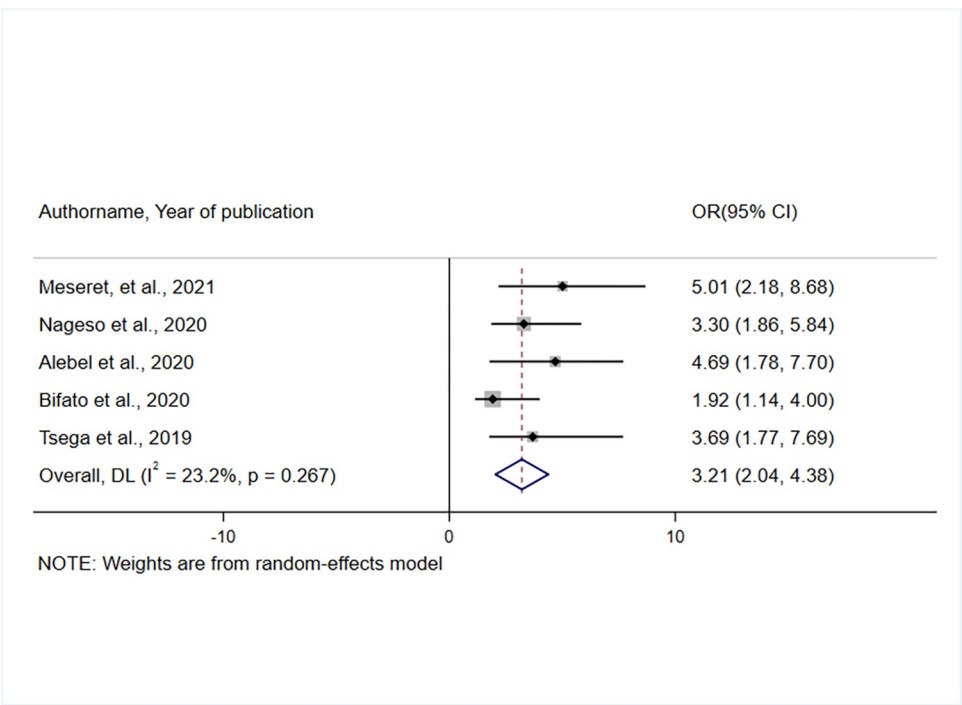

**Fig 9. Forest plot showing the association between perceived service quality and enrolment in the CBHI scheme in Ethiopia, 2017–2022.**

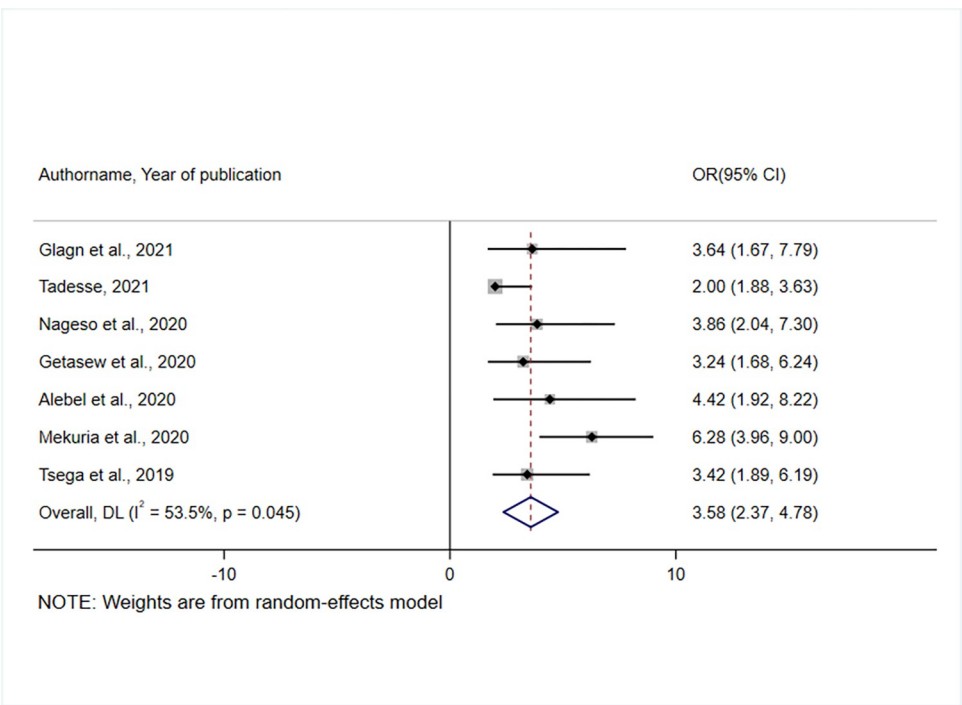

**Fig 10. A forest plot showing the association between having a person in the household with a chronic disease and enrollment in the CBHI scheme in Ethiopia, 2017–2022.**

The effect of having a person with chronic disease in the household was examined based on the results of seven studies [36, 38–40, 43–45]. As a result, households having a member with a chronic disease were 3.58 times more likely than their counterparts to join the CBHI scheme (OR = 3.58, 95% CI: 2.37, 4.78) (**Fig 10**). The heterogeneity test indicated the presence of heterogeneity, $I^2$ = 53.5%, hence the random effect model was assumed in the analysis [49].

The result of pooled estimates of four studies [39, 40, 46, 48] revealed an association between household trust in the CBHI scheme and enrolment status. Respondents who had trust in the CBHI scheme were 2.3 times more likely to enrol than those who did not (OR = 2.32, 95% CI: 1.57, 3.07). The heterogeneity test($I^2$ = 9.4%) indicated low variability, and hence the fixed effect model was assumed during the analysis [49, 50] (**Fig 11**).

## Discussion

A well-functioning and equitable healthcare system are vital for ending the vicious cycle of poverty and illness [51]. Enormous momentum has been gained after the WHO endorsed a resolution encouraging countries to advance toward UHC [3]. This change led to a shift away from OOP expenditures to enrollment in health insurance schemes like CBHI, ensuring individuals have affordable access to basic health interventions without the risk of getting impoverished [52].

Although the CBHI scheme protects members from catastrophic healthcare expenditure, its low enrolment and poor management limit its capacity to ensure UHC in a meaningful way [8]. Considering the significance of high enrolment for the viability of the CBHI scheme, assessing enrolment status at the national level allows for a better understanding of its potential contribution to UHC [11]. Hence, this systematic review and meta-analysis was aimed at determining the prevalence of CBHI enrolment and the factors that influence it in Ethiopia.

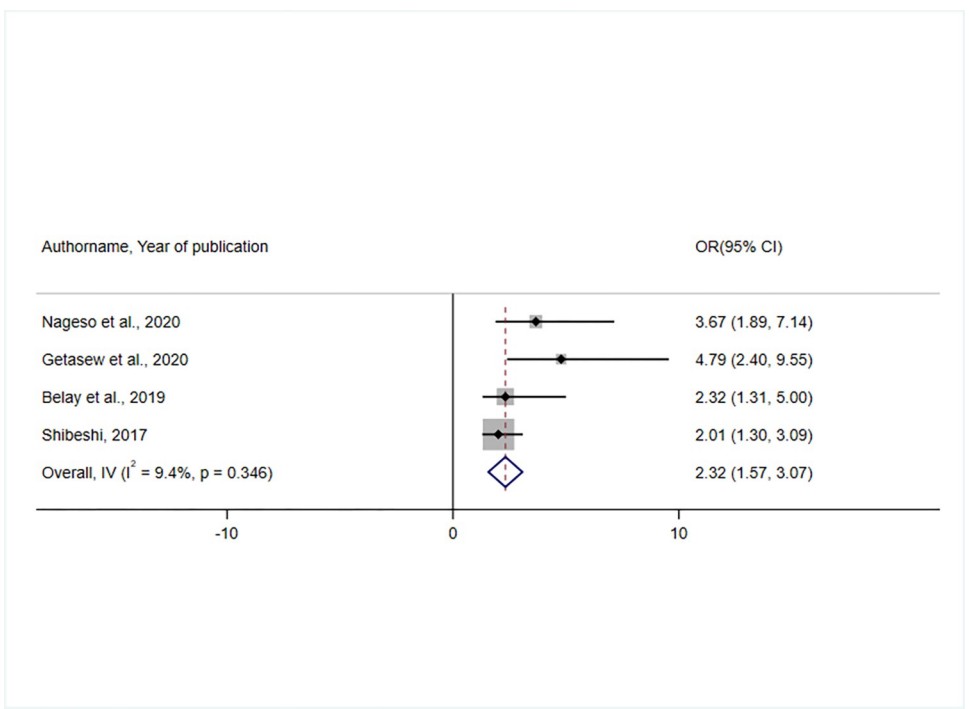

**Fig 11. A forest plot showing the association between having trust in the scheme and CBHI enrolment in Ethiopia, 2017–2022.**

In this analysis, the pooled prevalence of CBHI enrollment in Ethiopia was 44.5% (95% CI: 32.19, 58.50). Since no systematic reviews or meta-analyses have been conducted in Ethiopia, we have compared the current findings with other primary studies conducted outside of Ethiopia. Accordingly, the level of enrollment was comparable with studies conducted in Nigeria (48.4%) [53], Ghana (44%) [54], and Senegal (50%) [54]. On the other hand, enrollment was lower than in studies conducted in Rwanda(85%) [55], and Tanzania(59%) [56], and higher than in a study conducted in Kenya (10%) [57]. Differences in the knowledge of the community on the scheme, socioeconomic status, cultural issues, and a lack of managerial commitment at the lower levels might all have contributed to the disparity in prevalence. Another factor contributing to the disparity might be the current meta-analysis considered studies that only looked at active enrollment status (households who pay a premium for a complete year and possess an updated service card).

The pooled prevalence is too low compared to the aim of UHC, which is to offer health care to all citizens in a given country by mitigating the possible financial risk [58]. Furthermore, it was much lower than the Ethiopian government's goal of expanding CBHI schemes to 80% of districts and enrolling at least 80% of households by 2020, as stated in the health sector transformation plan (HSTP) [28]. Stakeholders at different administrative levels of government need to work together to create an accountable and transparent governance framework that fosters meaningful community participation to increase the enrolment rate [28]. Furthermore, officials who lead the CBHI scheme need to work to increase the enrolment rate by identifying bottlenecks and scaling up best practices to gain momentum in achieving UHC.

Based on the subgroup analysis, CBHI enrollment was highest in the Amhara region [68.03% (95% CI: 55.5, 80.5)], while it was lowest in the Sidama region [16.5% (95% CI:9.25,23,75)]. This disparity could be due to differences in socioeconomic status, and the

number of studies included in the meta-analysis. Furthermore, because the Sidama region is a newly emerged geographical region of Ethiopia as of November 23, 2021, there may be a gap in stringently implementing the program, resulting in low enrollment. Therefore, in regions with low enrollment rates, both upper and lower-level administrative bodies should engage in strategies to increase enrolment through community mobilization.

The affordability of premium payment for the scheme, respondents' knowledge of the CBHI scheme, perceived quality of service, trust in the scheme, and the presence of a person with a chronic disease in the household were all found to be determinants of CBHI enrolment.

The meta-analysis showed that knowledge of CBHI was found to influence the enrolment status. This was supported by studies conducted in Kenya [57, 59], Uganda [60], Cameroon [61], Nigeria [62, 63], and India [64] in which poor knowledge limited enrollment in the CBHI scheme. Individuals with more knowledge are more likely to have detailed information on the payment system, risk-pooling, financial resource redistribution, CBHI managerial structure, responsibilities of various managerial levels, and the benefits of being a member, all of which lead to them enrolling in the CBHI scheme [65]. In general, the concept of insurance and risk pooling is relatively new for many people in low-income countries including Ethiopia [10]. It is well known that having a basic understanding of insurance principles and their mechanisms has a positive influence on enrolment and renewal decisions. Hence, CHWs need to improve knowledge of CBHI in rural communities by providing information, education, and communication (IEC) on the related principles.

The results of the meta-analysis showed a positive association between the affordability of the premium with CBHI enrolment, which was supported by three other studies conducted in LMICs, including a systematic review and meta-analysis [1], a systematic review [21], and a review [66]. Primary studies conducted in Nepal, Senegal, and Ghana also supported this finding [67–69]. In Ethiopia, premiums for CBHI scheme membership were collected from households at a pre-determined fixed amount, which meant that every household paid the same amount of money without consideration of their socio-economic and livelihood status [70]. The majority of lower-income-generating households, particularly in rural areas, exhibit poor saving habits and may not have enough money on hand to pay premiums on time, which may have a significant impact on the enrolment rate. Strong emphasis should be placed on mitigating the difficulty experienced by members during premium payment by aligning payment time with their income source, such as during crop harvesting season. This challenge can be overcome by forming local solidarity groups, allowing payments in installments, and launching local initiatives to assist poor members, which might raise the number of enrollees in the scheme [21].

As governments strive to achieve "Health for All", it is critical to consider the quality of healthcare services [8]. In this systematic review and meta-analysis, the rate of CBHI enrolment was found to be influenced by the perceived quality of service provided at health facilities. This was supported by two systematic reviews and meta-analyses undertaken in LMICs [10, 21] and primary studies conducted in India [64], Kenya [57], Uganda [60], and Burkina Faso [71]. Enrollment in the CBHI program could likely be improved if people received high-quality care at health facilities during their initial visits, which is the primary objective of the CBHI scheme [72, 73]. Providing high-quality health services is crucial for achieving UHC at the individual and societal levels [3]. Thus, stakeholders in the health system need to focus on the dimensions of high-quality healthcare: effectiveness, safety, people-centeredness, timeliness, equity, integration, and efficiency [74]. To increase the enrolment and sustainability of a CBHI scheme, the technical competency of health providers, patient-provider relationships, and availability of medical supplies should all be improved.

In addition, this meta-analysis revealed that individuals living with chronic illnesses in the household influenced CBHI enrollment. Studies conducted in Burkina Faso [71], India [75], and Bangladesh supported this finding [76]. A systematic review and content analysis conducted in LMICs also supported the current finding [22]. These results suggest that relatively high-risk individuals could prefer to be insured to reduce financial catastrophes from OOP expenditures for repeated visits to health facilities for routine follow-up [77]. Moreover, this indicates an adverse selection problem, putting the long-term sustainability of the scheme in jeopardy. This might threaten risk-sharing principles, as people with chronic medical illnesses are more likely to get sick repeatedly, thus increasing the chances of exhausting all resources within a short time [78]. As a result, CBHI offices at all levels should consider implementing targeted subsidies for high-risk groups, with a robust plan to address the financial gap left by adverse selection. On the other hand, there is a need to provide high-risk groups with health promotion measures, such as physical activity, dietary modifications, family support, and care to limit the number of outpatient visits and hospital admissions leading to resource depletion.

Finally, the trustworthiness of the CBHI scheme by individuals was reported to be a facilitator of insurance enrolment, which was supported by two systematic reviews and meta-analyses and another study conducted in LMICs [10, 21] and a primary study conducted in Nigeria [79]. As a result, stakeholders at various levels of government must collaborate to build a responsible and transparent governance system that promotes the scheme's trustworthiness to boost enrolment rates. Furthermore, scheme administrators need to be more responsive to control and support the scheme in line with community preferences and general satisfaction to develop trust in the scheme and increase enrolment [21].

This systematic review and meta-analysis was the first of its kind in Ethiopia to assess the level of CBHI enrolment and its determinants, and it may provide valuable input to policymakers and managers at various levels on their road to achieving UHC by addressing the challenges of enrolment. However, the findings should be interpreted with caution due to some of the limitations listed below. First, only articles published in English were included in the search. The majority of studies considered were cross-sectional, making it difficult to establish a cause-effect relationship due to the nature of the study design. Furthermore, the studies were limited to five regions, which may limit the representativeness of the findings. Finally, due to a dearth of similar systematic reviews and meta-analyses, we were obliged to discuss some of our findings, particularly the enrolment status with primary studies conducted outside of Ethiopia.

## Conclusion

According to this meta-analysis, CBHI enrollment in Ethiopia was low (45.5%), compared to the HSTP goal of 80% by 2020 [28]. Affordability of the premium for the CBHI scheme, knowledge of respondents on the scheme, perceived quality of service, trust in the scheme, and the presence of individuals living with chronic diseases in the household were all found to influence CBHI enrolment. Community health workers (CHWs) need to focus on improving knowledge of CBHI in rural communities by providing health education. To deal with the issue of affordability, due emphasis should be placed on building local solidarity groups and strengthening local initiatives to aid poor members. Stakeholders in health service delivery need to emphasize the dimensions of high service quality. The financial gap created by the adverse selection of households with chronically ill members should be rectified by implementing targeted subsidies with robust plans. Strengthening community trust through transparent management of the scheme should be encouraged.

## Supporting information

**S1 File. PRISMA checklist 2020 used to report the result of systematic review and meta-analysis.**
(DOCX)

**S2 File. Examples of the search strategy for systematic review and meta-analysis on the level of enrolment in the CBHI scheme and its determinants in Ethiopia, 2022.**
(DOCX)

**S3 File. JBI critical appraisal checklist for prevalence and case-control studies used for assessing the individual quality of all studies included in the systematic review and meta-analysis, 2022.**
(DOCX)

**S4 File. List of variables considered for estimation of pooled odds ratio.**
(XLSX)

**S5 File. Minimal data set that is used to estimate the pooled prevalence.**
(DTA)

## Acknowledgments

We thank Wachemo University, College of Medicine and Health Sciences for providing us with full internet access while we were working on this study. We would like to thank all authors of the studies included in this systematic review and meta-analysis.

## Author Contributions

**Conceptualization:** Aklilu Habte, Samuel Dessu.

**Data curation:** Aklilu Habte, Fitsum Endale.

**Formal analysis:** Aklilu Habte, Aiggan Tamene.

**Investigation:** Tekle Ejajo, Samuel Dessu.

**Methodology:** Aklilu Habte, Aiggan Tamene, Tekle Ejajo, Samuel Dessu, Addisalem Gizachew, Dawit Sulamo.

**Project administration:** Aklilu Habte.

**Software:** Aklilu Habte, Aiggan Tamene.

**Supervision:** Aklilu Habte, Fitsum Endale, Dawit Sulamo.

**Validation:** Aklilu Habte, Tekle Ejajo.

**Visualization:** Aklilu Habte.

**Writing – original draft:** Aklilu Habte, Aiggan Tamene, Tekle Ejajo, Fitsum Endale.

**Writing – review & editing:** Aklilu Habte, Aiggan Tamene, Samuel Dessu, Addisalem Gizachew, Dawit Sulamo.

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
