## [Decision Letter · Decision Letter 0]

16 Jun 2022

PONE-D-22-10028Towards Universal Health Coverage: The level and determinants of  enrolment in Community Based Health Insurance(CBHI) scheme in Ethiopia: Systematic Review and Meta-analysisPLOS ONE

Dear Dr. Habte,

Thank you for submitting your manuscript to PLOS ONE. After careful consideration, we feel that it has merit but does not fully meet PLOS ONE’s publication criteria as it currently stands. Therefore, we invite you to submit a revised version of the manuscript that addresses the points raised during the review process.

 At the outset, I would like to congratulate authors for highlighting the  issue of UHC and its determinants with in the context of Ethiopia's CBHI. Simultaneously, I would like to specially acknowledge the contributions of both reviewers for submitting  exceptional and exhaustive reviews. It would definitely add to the value of the contents. Henceforth,  authors are humbly requested to kindly consider/relook/rebutt the various minor revision points as raised by the reviewers. This would further enhance the scientific validity of the manuscript and will definitely improve the quality of the study. Suggestions proposed by one reviewer about publication bias  need revisit. Please look into grammatical and typo errors specifically before submitting revised version. Please have a relook through track changes word document or seek help of the journal supportive staff if needed. I am looking forward for re-submission so that the findings can be disseminated to the scientific community at earliest possible and feasible. Thanks 

We look forward to receiving your revised manuscript.

Kind regards,

Gopal Ashish Sharma, MBBS, MD

Academic Editor

PLOS ONE

Journal Requirements:

3. Please ensure that you refer to Figure 10 in your text as, if accepted, production will need this reference to link the reader to the figure.

4. We note that this manuscript is a systematic review or meta-analysis; our author guidelines therefore require that you use PRISMA guidance to help improve reporting quality of this type of study. Please upload copies of the completed PRISMA checklist as Supporting Information with a file name “PRISMA checklist”.

Reviewers' comments:

Reviewer's Responses to Questions

**Comments to the Author**

1. Is the manuscript technically sound, and do the data support the conclusions?

Reviewer #1: Yes

Reviewer #2: Partly

2. Has the statistical analysis been performed appropriately and rigorously? 

Reviewer #1: Yes

Reviewer #2: Yes

3. Have the authors made all data underlying the findings in their manuscript fully available?

Reviewer #1: Yes

Reviewer #2: No

4. Is the manuscript presented in an intelligible fashion and written in standard English?

Reviewer #1: Yes

Reviewer #2: No

5. Review Comments to the Author

Reviewer #1: Dear Editor: Dr. Gopal Ashish Sharma

Manuscript Number: PONE-D-22-10028

Manuscript Title: Towards Universal Health Coverage: The level and determinants of enrolment in Community Based Health Insurance (CBHI) scheme in Ethiopia: Systematic Review and Meta-analysis

I read the manuscript completely and I found that the authors have done remarkable research on the level and determinants of enrolment in the Community Based Health Insurance (CBHI) scheme in Ethiopia. I think this research could provide health policymakers with sound evidence to overcome the challenge of achieving Universal Health Coverage in the Ethiopian health system. Using a systematic review and meta-analysis method for answering the research question is a good option that could provide the needed evidence. Also, in this study, statistics, and other analyses are performed to a high technical standard and are described in sufficient detail. However, the manuscript needs minor improvements that could have a better presentation if they are considered by the authors.

Abstract:

- Google Scholar is a search engine and not a database. Please correct.

- In the abstract section. It is best to draw a clearer picture of the study. If you started in a structured way, be sure to specify the other sections as well.

- Try to make the introduction part of the abstract unique and not copy-paste from the manuscript.

Introduction section:

- Please use the following articles in this section of the introduction:

DOI: https://doi.org/10.2147/CEOR.S254946

DOI: https://doi.org/10.1186/s12913-021-06673-0

DOI: 10.47176/mjiri.35.191

Discussion section:

- All abbreviations were defined when used for the first time in the manuscript: such as (LMIC)

Best regard

Naser Derakhshani

Ph.D. in Health Service Administration

Reviewer #2: Summary

Overall, this paper provides an interesting systematic review and meta-analysis that presents a pooled prevalence on the CBHI enrolment in Ethiopia. The answer to question 3 was provided as it is unclear if the study protocol was published on any platform, such as PROSPERO. Otherwise all data seems to have been made available. In addition, the response to question 4 was given since the manuscript would benefit from further corrections as in the comments for consideration listed below and the attached Word document.

Revision comments or further considerations:

1) Recommend some revisions to the introduction to facilitate readership and add clarity. Particularly, further clarification to the reader as to why certain countries were selected as a comparison in the first paragraph (lines 81-82) and why UHC would be assured through UHC, as stated in line 158. This notion reappears in lines 504-505, which implies that a country would solely require CBHI in order to reach UHC, which is quite a bold statement. The scope of the CBHI scheme can be quite limiting, not all-encompassing and can face sustainability issues, thus stating the prevalence of CBHI enrolment in this way seems reductionist. If this was not intended, then it could help to slightly modify these sentences, so as to not be too limiting.

2) For the study design (lines 196-201), the study characteristics for PICOS are referred to in the S1File, i.e. the PRISMA guideline checklist. However, this is not clearly indicated within the methods section. Could this be clarified or incorporated?

3) Table 2 of the supplementary file 3 “S3File”, the critical appraisal tool, does not have any values within the tenth column. Was this intended or missed? Would recommend clarification.

4) In terms of the pooled prevalence, specifically line 456, it would be helpful to know what happened to the other four studies. Was no enrolment prevalence or number of people enrolled in the scheme reported? Would suggest describing this outcome for the four studies in the previous section called “Characteristics of included studies”. This way the reader does not have to examine the table and think of reasons why or be left guessing.

5) For Figure 4 (descriptions mentioned in lines 481-483), why does the grouping called “Since 2020” in the forest plot also include one study from 2020 (Alebel et al. 2020) and one from 2019 (Tsega et al. 2019), when those would actually make up the “Before 2020” category or grouping? Was this intentional?

6) For line 466, why is Figure 4 showing a pooled estimate of 45.50% and a 95% confidence interval of 32.19 to 58.82 instead of the main reported finding?

7) The manuscript briefly touches on stakeholders (e.g. lines 508-510 and 585-587), is there further information on the type and role of stakeholders involved in carrying the CBHI scheme forward and sustaining its implementation? It would be helpful to know more about the involvement and possible contribution of other stakeholders for enrolment of the scheme.

8) The figures and tables require some standardisation, especially in terms of the units and column headings, which either are not uniform or as clear as they could be. Would recommend another look to refine these.

9) It seems that the study may indicate publication bias, as the lines 301-303 state: “Statistically, Begg's and Egger's tests were conducted, and a p-value < 0.05 indicated the presence of a small study effect (publication bias)” and the funnel plot (Figure 5) shows minor asymmetry towards the right side. If so, this would need to be reflected within the limitations section and the authors would need to discuss the implications of this publication bias.

10) The abstract is missing the heading “Conclusion”. Is the highlighted text within the attached Word document the conclusion?

11) Minor edits to certain tables and figures are needed to improve consistency and comprehensibility to the reader. In addition, the supplementary files should also include figure titles. Supplementary Figure 1 has a few minor errors in the text, which would benefit from correction (e.g. “define” instead of “defined”, etc).

12) Some of these formatting issues have been amended using tracked changes in the Word document. However, another look would be required prior to finalisation to improve the English language and consistency of spelling throughout.

13) A few comments have been included in the Word document for further clarification.

14) The references section requires another review to correct the inaccurately listed authors, specifically publications from organisations and agencies. Only some of these inaccurate references have been highlighted in the attached Word document.

15) Finally, was limiting the scope of eligible studies to English only an issue to consider in the limitations section of the discussion? Could any studies potentially have been missed?

6. PLOS authors have the option to publish the peer review history of their article (what does this mean?). If published, this will include your full peer review and any attached files.

Reviewer #1: No

Reviewer #2: No

---

## [Author Response · Author response to Decision Letter 0]

21 Jun 2022

The responses to editors and reviewers have been attached as a "Response to Reviewers" in the submission system

---

## [Decision Letter · Decision Letter 1]

1 Aug 2022

Towards Universal Health Coverage: The level and determinants of enrollment in the Community-Based Health Insurance (CBHI) Scheme in Ethiopia: A Systematic Review and Meta-analysis

PONE-D-22-10028R1

Dear Dr. Habte,

We’re pleased to inform you that your manuscript has been judged scientifically suitable for publication and will be formally accepted for publication once it meets all outstanding technical requirements.

Kind regards,

Gopal Ashish Sharma, MBBS, MD

Academic Editor

PLOS ONE

Additional Editor Comments (optional):

Reviewers' comments:

Reviewer's Responses to Questions

**Comments to the Author**

1. If the authors have adequately addressed your comments raised in a previous round of review and you feel that this manuscript is now acceptable for publication, you may indicate that here to bypass the “Comments to the Author” section, enter your conflict of interest statement in the “Confidential to Editor” section, and submit your "Accept" recommendation.

Reviewer #1: All comments have been addressed

Reviewer #2: (No Response)

2. Is the manuscript technically sound, and do the data support the conclusions?

Reviewer #1: Yes

Reviewer #2: Yes

3. Has the statistical analysis been performed appropriately and rigorously? 

Reviewer #1: Yes

Reviewer #2: Yes

4. Have the authors made all data underlying the findings in their manuscript fully available?

Reviewer #1: Yes

Reviewer #2: Yes

5. Is the manuscript presented in an intelligible fashion and written in standard English?

Reviewer #1: Yes

Reviewer #2: Yes

6. Review Comments to the Author

Reviewer #1: I read the revised article, it is a good job and it can be of interest to countries for policy making in the field of universal health coverage.

best regard

Reviewer #2: Happy to recommend for publishing following the incorporation of some final minor comments below.

Minor changes:

- Line 57: In the abstract, the reported confidence interval for the pooled prevalence does not fully match. It should be adjusted to the reported interval of 95% CI: 32.19, 58.82 (as stated on page 10, line 272).

- Line 284: Minor issue but important. Please correct the typo of the comma for the confidence interval value 23.75. It should read as 23.75 not 23,75.

- Line 191: “PRISMA reporting guideline for PICOS”: would remove ‘for PICOS’ as this addition does not make sense. The guidelines were followed as a whole. Within the attached PRISMA checklist, it states PICOS were mentioned on page 4 of the manuscript, which presumably was chosen by the authors not to be disclosed beyond the eligibility criteria, which is fine.

- Line 356: “is vital” instead of “are”. Thus, “A well-functioning and equitable healthcare system is vital for…”. Unless the author had intended to use the plural form, in which case it would read as “systems… are”, therefore: “Well-functioning and equitable healthcare systems are vital for…”

No further review needed.

7. PLOS authors have the option to publish the peer review history of their article (what does this mean?). If published, this will include your full peer review and any attached files.

Reviewer #1: No

Reviewer #2: No

---

## [Editor Report · Acceptance letter]

3 Aug 2022

PONE-D-22-10028R1 

Towards Universal Health Coverage: The level and determinants of enrollment in the Community-Based Health Insurance (CBHI) Scheme in Ethiopia: A Systematic Review and Meta-analysis 

Dear Dr. Habte:

I'm pleased to inform you that your manuscript has been deemed suitable for publication in PLOS ONE. Congratulations! Your manuscript is now with our production department. 

Kind regards, 

on behalf of

Dr. Gopal Ashish Sharma 

Academic Editor

PLOS ONE